# Phytogenic Bioactive Compounds in the Diet of Lactating Sows, Litter Performance, and Milk Characteristics

**DOI:** 10.3390/ani13172764

**Published:** 2023-08-30

**Authors:** Gleyson Araújo dos Santos, Maria do Carmo de Oliveira, Amanda Medeiros Araújo de Oliveira, Victor Hugo Teixeira Batista, Priscila Oliveira Costa, Augusto Heck, Camilla Mendonça Silva, Adriano Henrique do Nascimento Rangel, Michelly Fernandes de Macedo, Rennan Herculano Rufino Moreira

**Affiliations:** 1Department of Animal Science, Universidade Federal Rural do Semi-Árido, Mossoró 59625-900, RN, Brazilpocosta10@gmail.com (P.O.C.);; 2Department of Animal Science, Universidade Federal do Ceará, Fortaleza 60356-001, CE, Brazil; amanda.oliveira@reginaalimentos.com.br; 3Department of Veterinary College, Universidade Federal do Rio Grande do Sul, Porto Alegre 90040-060, RS, Brazil; augusto.heck@dsm.com; 4Academic Unit Specialized in Agricultural Sciences, Universidade Federal do Rio Grande do Norte, Macaíba 59280-000, RN, Brazil

**Keywords:** plant extracts, essential oil, piglets, maternity, pig farming

## Abstract

**Simple Summary:**

Lactating hyperprolific sows present with oxidative stress, which affects feed intake and milk production. The metabolic response of animals is to mobilize body reserves, resulting in a negative energy balance. In this context, this study suggested the use of phytogenic additives in the feed of lactating sows to improve body conditions during the lactation period and milk production, yielding adequate piglet performance. The study verified that using phytogenics in the feed of lactating sows resulted in improved bromatological parameters of the milk and improved the performance of sows and piglets.

**Abstract:**

The objective was to evaluate the effect of phytogenic supplementation in the feed of lactating sows on the performance parameters of sows and suckling piglets. Ninety-three lactating sows of commercial lines (sows TN70) were distributed in a completely randomized design. The treatments adopted were a control diet without phytogenics (control) and a control diet with phytogenic supplementation in the feed. Performance parameters and the behaviors of sows and piglets were evaluated, in addition to milk composition and the biochemical parameters of the animals. The use of phytogenics did not affect the feed intake or tissue mobilization of the sows. However, it improved the production and quality of milk and reduced the possibility of light piglets at weaning by 22.16 percentage points. Regarding biochemical parameters, phytogenics improved animal recovery in the lactation period, as proven by an increase in the serum concentration of total protein and urea. Sows that received phytogenics exhibited increased behaviors of inactivity (3.16%), breastfeeding (1.15%), and water consumption (0.95%). In conclusion, using phytogenics in the feed of lactating sows improves the performance of the litters without affecting the body condition of the sows, with increased milk production and enhanced protein and lactose concentration.

## 1. Introduction

Modern sow lineages produce a large number of piglets born with a high capacity for lean tissue deposition [1]. These females require high nutritional intake due to increased milk production, and because voluntary intake is generally insufficient, the sows metabolize body reserves, culminating in a negative energy balance [2]. This imbalance increases the number of non-productive days in females, which often exhibit high weight loss during lactation and reduced milk production due to low feed intake, affecting litter development and the following reproductive cycle [3].

Furthermore, it has been reported that hyperprolific lactating sows present oxidative stress, which affects feed intake and milk production [4]. Oxidative stress can generate changes in the behavior of sows; Shein [5] observed that sows with a high intensity of oxidative stress frequently moved around after farrowing, potentially crushing and causing deaths of piglets.

Thus, to improve the intestinal health of sows, increase feed intake during the lactation phase, and improve litter performance, natural compounds called phytogenics have been investigated; these are products based mainly on plant extracts, such as essential oils, spices, and organic acids [6,7,8].

Phytogenics are mixtures of volatile, lipophilic, and low-molecular-weight substances that are usually odorous and in liquid form and have active compounds of cinnamaldehyde, carvacrol, thymol, anethole, and limonene [9]. These components exhibit antimicrobial, anti-inflammatory, and antioxidant actions and modulate the intestinal microbiome [10]. They also stimulate digestive secretions and immune stimulation [11]. Wang et al. [7] verified that supplementation with phytogenics of mixed herbal extracts in sows during late pregnancy and lactation enhanced the litter weight gain and average daily feed intake of sows during lactation, while decreasing diarrhea in suckling piglets. On the other hand, Reyes-Camacho et al. [6] observed that, despite not changing the feed intake of the sows, the components of the phytogenic compound were transferred to the milk, resulting in improved intestinal health in piglets after weaning.

Given this information, we hypothesized that using phytogenics in the feed of sows in the lactation phase would provide better female body conditions during the lactation period, better litter performance, and improved behavior. Thus, this study aimed to evaluate the effect of phytogenic supplementation in the feed of lactating sows on the performance parameters of lactating females and suckling piglets. 

## 2. Materials and Methods

The Ethics Committee for Research on Animal Production of the Federal Rural University of the Semi-Arid Region approved all experimental procedures under protocol No. 01/2022.

### 2.1. Experimental Design

The experiment was conducted on a commercial farm with 93 lactating sows of different farrowing orders (mean parity 4.62 ± 1.45) from a hyper-proliferous commercial lineage (line TN70). The females were housed in individual farrowing stalls (0.6 × 2.2 m) with lateral areas (0.6 × 2.2 m) exclusive to piglets. In the farrowing stalls, two-thirds of the floor was slatted; the environment was air-conditioned; and the stalls did not contain enrichment materials.

The internal environment of the farrowing stalls was characterized by measuring the ambient temperature and relative humidity using dataloggers configured for data collection at five-minute intervals. The dataloggers were installed at half the height of the sows. 

The sows were distributed in a completely randomized design consisting of two experimental treatments (control diet *n* = 49; control diet + phytogenic compounds *n* = 44). One sow and its litter represented one experimental unit. The distribution of the sows in the treatments considered the body weight and the order of farrowing.

The control diet was based on corn and soybean meal (Table 1), according to the nutritional requirements of lactating sows reported in the lineage technical manual [12]. All diets were provided in mash form.

The phytogenic additive (Digestarom^®^ Sow, DSM Nutritional Products, São Paulo, Brazil) was added and mixed with the control diets, at the time of the daily meal, according to the intake of sows and following the manufacturer’s recommendation of a 150 g/ton ratio.

The phytogenic additive contained a proprietary mix of based on essential oils, herbs, and extracts, including caraway essential oil, thyme essential oil, anise essential oil, orange essential oil, mint essential oil, and licorice extract; the major essential oils were menthol (1.8 mg/kg), trans-anethole (0.76 mg/kg), and thymol (0.41 mg/kg) [13].

The sows began supplementation on the second day of lactation (0.15 g/kg of feed). The sows were fed 2.0 kg of experimental diets during lactation until farrowing. After farrowing, 1.0 kg was offered; on the second day, 2.0 kg; on the third day, 3.0 kg; on the fourth day, 4.0 kg; on the fifth day, 5.0 kg; on the sixth day, 6.0 kg; on the seventh day, 7.0 kg; and from the eighth day until weaning, 8.0 kg of feed. The sows received water ad libitum, and the feeding was divided into five feed offerings per day during the lactation period.

### 2.2. Biochemical Blood Parameters

The blood of 13 females per treatment was collected on the 2nd day of lactation and at weaning to measure serum concentrations of urea, total protein (TP), albumin (ALB), aspartate aminotransferase (AST), glutamyl aminotransferase (GGT), creatine kinase, triglycerides (TG), total cholesterol, and glucose (GLU) using a semi-automatic biochemical analyzer (model BIO-2000^©^, Bioplus Instruments, São Paulo, Brazil), through corresponding commercial kits. 

### 2.3. Bromatological Composition of the Milk

The bromatological composition of the milk was evaluated using 13 females per treatment (*n* = 26) selected based on their weight, on the second day after farrowing (equalization), and at weaning. Milk ejection was induced by applying 1 mL of intravenous oxytocin in the marginal ear vein of the sows, which were fasted for 1 h. Subsequently, 60 mL of milk was collected from each animal which was immediately identified and stored at −20 °C [14].

The total crude protein, fat, lactose, degreased dry extract, and solid milk samples were determined by infrared absorption (Bentley 2000^®^, Bentley Instruments Inc., Chaska, MN, USA). 

### 2.4. Sow Behavior

Sow behavior was monitored every ten minutes for 24 h, beginning at 06:00 on the 7th and 15th days of lactation. The following behavioral activities were observed: drinking water (DW), feed intake (FI), stereotyped, agonistic behavior (S), inactive (I), inactive alert (IA), breastfeeding (B), biting (BT), and poking (P), as suggested by Pandorfi et al. [15]. 

### 2.5. Sow Performance

All sows were weighed after farrowing (equalization) and weaning to verify body mobilization. The daily feed intake of the females and milk production were evaluated during the entire lactating period. Milk production was estimated using the equation suggested by Noblet and Etianne [16]: milk production (kg/day) = {(0.718 × daily weight gain of the piglet (g) − 4.9) × number of piglets}/0.19.

### 2.6. Parameters Evaluated in the Litter

The litter was equalized for 13 or 14 piglets one day after farrowing. The piglets in each litter were identified and weighed to determine the weight gain of the period one day after farrowing and weaning. Mortality was measured. 

Stratification of the body weight of the litters was measured from the percentile of the individual weights of the piglets at weaning in each litter, which were distributed in three classes: light, medium, and heavy. Litter equalization adopted the stratification for the body weight of piglets with >1.4 kg, corresponding to light piglets; 1.4 to 1.7 kg, corresponding to medium piglets; and <1.7 kg, corresponding to heavy piglets. The stratification adopted for weaning was: >4.9 kg for light piglets, 4.9 to 6.1 kg for medium piglets, and <6.1 kg for heavy piglets.

The frequency of diarrhea was evaluated during the lactation period using the methodology suggested by Gonçalves et al. [17]. To determine the stool consistency score, a visual evaluation was performed daily, morning and afternoon, with scores ranging from 0 to 3 for each animal: 1 = solid stools; 2 = pasty stools; and 3 = liquid stools. Scores of 2 and 3 indicated the occurrence of diarrhea. Thus, it was possible to calculate the frequency of days with the occurrence of diarrhea in each evaluation.

### 2.7. Piglet Behavioral Parameters

The behavior of the litter was monitored for 24 h at 7 and 15 days of lactation. The number, interval, and duration of sucklings were evaluated when 50% + 1 of the piglets of the litter began suckling, and ended when more than half of the litter left the teats or presented inactive behavior, as recommended by Moreira et al. [18].

### 2.8. Statistical Analysis

SAS (9.3) software [19] was used for statistical analyses. The data were subjected to Shapiro–Wilk tests at a 5% probability level to verify data normality. 

Analysis of variance (ANOVA) was performed, and means were compared using F-tests, considering a significant effect less than or equal to 5% probability and a tendency between 5% and 10%. 

The data that did not present normal distribution were normalized by the PROC RANK in the SAS (9.3) software [18]. The Kruskal–Wallis test was used to compare non-normalized data at a 5% probability.

The following model was used:$$Y_{ijk}=µ+{£}_{i}+\beta_{i}+{Ǥ}_{i}+\varepsilon_{ijk}$$
where Y_ijk_ is the observation of the effect of supplementation with phytogenic i, replicate j, and experiment k; μ is the overall mean; £i is the random effect of farrowing order; β_i_ is the random shed effect; Ǥ_j_ is the fixed effect of phytogenic supplementation; and ε_ij_ is the random error associated with each observation, considering independence, identical distributions, normal, mean 0, and variance σ.

## 3. Results

### 3.1. Environmental Characterization 

The temperatures observed during the experimental period in the maternity wards are shown in Figure 1.

The thermal environment was within the limits for the category. Thus, the animals were mostly in the thermal comfort zone, about 67% of the day.

### 3.2. Performance of Sows Supplemented or Not with Phytogenics

Supplementation with phytogenics compared with the control diet did not affect (*p* > 0.05) the weight of sows during lactation or changes in feed intake. However, the sows that received the phytogenic compounds exhibited increased (*p* < 0.01) milk production by 1.71 kg/day (Table 2) compared with the control diet.

### 3.3. Bromatological Composition of the Milk of Sows Supplemented or Not with Phytogenics

The composition of milk in equalization did not differ (*p* > 0.05) between treatments, except for crude protein and lactose, which showed lower concentrations (*p* < 0.05) in treatments with phytogenic supplementation, of 0.25 and 0.39 percentage points, respectively (Table 3).

At weaning, there was a tendency for crude protein appearing in the milk of sows treated with phytogenics (*p* < 0.075) with a superiority of 0.15 percentage points (Table 3) compared with the control diet.

### 3.4. Biochemical Blood Parameters of Sows Supplemented or Not with Phytogenics

No differences (*p* > 0.05) for biochemical parameters among the evaluated experimental treatments related to the moment of equalization were observed. Th is also occurred during weaning, except in females supplemented with phytogenics, which increased (*p* < 0.05) the total protein and urea concentrations by 0.31 and 7.58 g/dL, respectively (Table 4).

### 3.5. Sows Behavior Supplemented or Not with Phytogenics

There was no effect (*p* > 0.05) on the analyzed behavioral variables, except for water intake, inactive behavior, and breastfeeding, which increased by 0.95, 3.16, and 1.15 percentage points, respectively, and a reduction (*p* < 0.05) of 6.22 percentage points for inactive alert behavior in treatments with phytogenic supplementation when compared with the control treatment (Table 5). The behavior was characterized as inactive when the sows were fully lying (ventral or laterally) with their eyes closed and without oral activity, while the inactive alert behavior was when the females were standing still, lying down, or sitting with their eyes open.

### 3.6. Litter Performance Depending on the Supplementation or Not with the Phytogenics

The parameters related to the moment of equalization did not differ (*p* > 0.05) between treatments (Table 6); there was also no difference (*p* > 0.05) between weight classes (light, medium, and heavy) in the equalization (Figure 2).

At weaning, the average litter weight, average piglet weight, and daily weight gain were higher (*p* < 0.01) when the sows were supplemented with 10.22, 0.767, and 0.035 kg phytogenic compounds, respectively, compared with the control diet. The coefficient of variation improved (*p* < 0.05) by 3.43 percentage points with supplementation. The use of phytogenics reduced (*p* < 0.05) the appearance of diarrhea, with greater severity (type 3), i.e., by 4.52 percentage points (Table 6), compared with the control diet.

In addition, phytogenic supplementation reduced (*p* < 0.05) the percentage of light piglets, those smaller than 4.90 kg at weaning, as well as increased (*p* < 0.05) the percentage of heavy piglets, those larger than 6.10 kg (Figure 2).

Supplementation with phytogenics in the feed increased the suckling duration (*p* < 0.05) by 1.67 min compared with the control diet. Additionally, the piglets suckled up to 2.29 times more per 24 h in this treatment. In addition, the interval between suckling reduced (*p* < 0.05) by 4.66 min (Table 6).

## 4. Discussion

The use of phytogenics did not interfere with the sows’ body mobilization and feed intake. These results are satisfactory because the sows that consumed the phytogenic presented an increase of 1.71 kg/day in milk production, i.e., even with the increase in milk production, there was no catabolism in the females. Notably, sows consume high levels of energy to maintain and produce milk during lactation. 

Therefore, the loss of body weight is inevitable. In the present study, a mobilization of body tissue of 2.29% was observed in the group that received the phytogenic compounds, equivalent to a 7 kg loss of body weight during lactation. According to Domingos et al. [20], a reasonable weight loss in lactation should not exceed 10 kg of body weight. 

The bromatological composition of the sow’s milk collected on the second day of lactation (equalization), when supplementation with the additive was started, differed only for the concentration of crude protein and lactase. There was also a tendency in total solid concentrations for treatment with phytogenics. In the present study, the females began supplementation on the same day of milk collection. Therefore, these results may be inherent to the animal characteristics, and not the experimental treatment.

Sows receiving the treatment of phytogenic compounds presented higher crude protein levels in the milk collected at weaning. This response is associated with the highest serum concentrations of total proteins observed in the present study for animals that received the phytogenic compounds. Adequate protein intake likely resulted in better milk quality, body condition maintenance, and greater circulation of serum urea, a marker of hepatic protein synthesis. 

Supplementation enables full recovery of these sows and maintains lactation without catabolism or mobilization of muscle tissue. If this mobilization had occurred, we would have altered markers of muscle injury (creatine kinase) [21]. 

Regarding total protein concentrations, Jang et al. [22] stated that most of the body’s protein synthesis occurs in the liver. This synthesis is directly related to the nutritional and metabolic status of sows. The better absorption of food provides more amino acids, which results in increased circulating proteins. 

The improvement in milk production influenced the average weight of the piglets at weaning and reduced the coefficient of variation of the litters from the supplemented females. Similar results were obtained by Nowland et al. [23], who supplemented hyper prolific sows in pregnancy and lactation with bioactive phytogenics, resembling the present study. They verified that the phytogenic compounds did not affect the sow’s feed intake and body condition, but improved piglet growth until weaning. 

The phytogenics adopted in this study were composed of essential oils and spices with antimicrobial and antioxidant actions because they are rich in carvacrol, thymol, anethole, and limonene [9]. These bioactive compounds act on intestinal health and enable greater energy efficiency by reducing the incidence of stressors [24], consequently improving milk production [7]. This condition was verified in the present study, where the increase in milk production of the sows resulted in piglets with enhanced growth during lactation and greater final weight at weaning (Figure 2). 

The present study showed that supplementation with phytogenic compounds improved litter uniformity, reducing the number of light piglets (less than 4.9 kg) by 22.16 percentage points and increasing the number of heavy piglets (greater than 6.1 kg) by 18.11 percentage points. This was likely due to the higher milk production of the females of this experimental group and the milk quality. Costermans et al. [25] suggest that piglet growth is associated with the bromatological composition of the milk; they observed a reduction in litter weight gain when the fat and protein concentrations in milk were lower.

In addition, it was found that the phytogenic compounds reduced the occurrence of severe diarrhea in piglets during the lactation phase. There was possibly a maternal transfer of the bioactive principles of the phytogenic compounds via milk, demonstrating antimicrobial action [6,23]. Reyes-Camacho et al. [6] identified high concentrations of thymol, anethole, and p-cymene in the milk of sows supplemented with phytogenic compounds rich in these components during lactation, inferring that there was a maternal transfer through the milk. There is evidence that the supply of intestinal- health-improving assets to lactating sows can manipulate the intestinal microbiota of the litter for up to two weeks after weaning [26]. The result was also observed by Nowland et al. [23], who identified a reduction in the occurrence of diarrhea in piglets from females supplemented with oregano essential oil.

The present study evaluated the effect of the supplementation of phytogenic additives on the behavior of the sows and on the suckling time of the piglets, in which the supplemented sows generally increased water intake and presented higher alert levels, possibly associated with the longer period available for feeding the litter. These behaviors are associated with the increased milk production of sows which received the bioactive compounds, justifying the reported behavioral conditions. It was also observed that the piglets of litters treated with phytogenic compounds increased their breastfeeding period, with longer durations of and shorter intervals between feedings. 

According to Moreira et al. [18], the appropriate feeding behavior for suckling piglets is directly related to higher milk production, consequently providing a good litter performance. The literature shows that the average duration of suckling piglets is 6 min [27,28]. These values are lower than those found in the present study, which was 9.37 min for litters from the group supplemented with phytogenic compounds and 7.70 min from the control group. These data indicate the comfort and welfare of the animals during lactation [18].

## 5. Conclusions

The phytogenic compounds yielded greater milk production, maintained the body condition of sows stable, and improved the performance of the litters until weaning.

## Figures and Tables

**Figure 1 animals-13-02764-f001:**
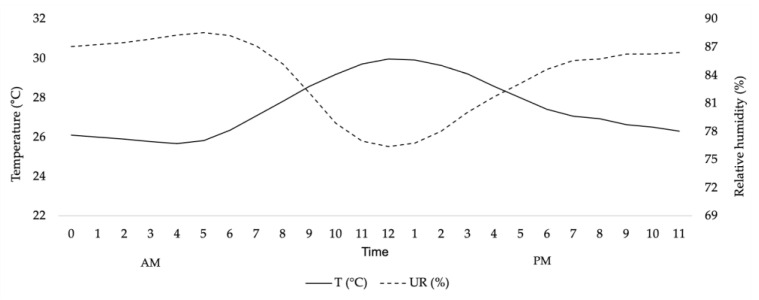
Average temperature and relative humidity of the maternity ward during the lactation (21 days) as a function the hours.

**Figure 2 animals-13-02764-f002:**
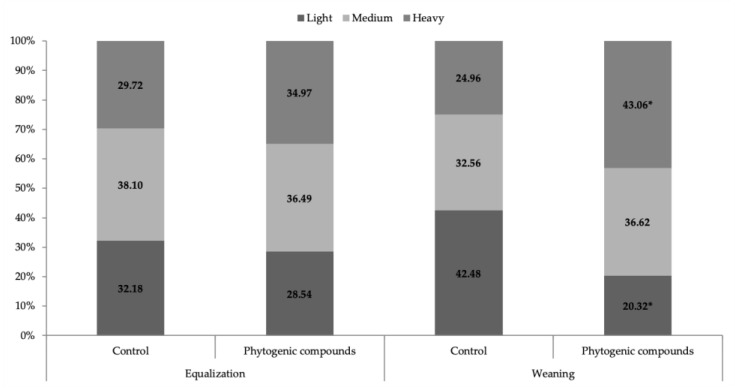
Stratification of piglet weight (light, medium, and heavy) as a function of supplementation with phytogenics (*n* = 44) or no supplementation with phytogenics, and a diet control of lactating sows (*n* = 49). We considered light piglets at equalization > 1.4 kg; medium piglets at equalization from 1.4 to 1.7 kg; and heavy piglets at equalization < 1.7 kg. We considered light piglets at weaning with >4.9 kg; medium piglets at weaning with 4.9 to 6.1 kg; and heavy piglets at weaning with <6.1 kg. * The means differed at the 5% probability level, confirmed by F-test.

**Table 1 animals-13-02764-t001:** Components of the experimental diet for sows in lactation.

Ingredients (%)	Lactation
Corn	54.43
Soybean meal	32.10
Wheat middlings	2.00
Lignocellulose ^†^	0.20
Sugar	3.00
Soybean oil	3.80
Vitamin D 250 (25OHD3) ^‡^	0.03
Vitamin and Mineral premix ^‡‡^	4.00
Mycotoxin adsorbent	0.10
DL-Methionine	0.05
Lysine 70% ^§^	0.09
Guanidinoacetic acid (GAA 96%) ^¶^	0.10
2% Biotin	0.01
Probiotic ^†††^	0.01
Probiotic ^¶¶^	0.10
Total	100.00
Nutrients	
Metabolizable energy (kcal/kg)	3450
Crude protein (%)	19.81
Calcium (%)	0.95
Available phosphorus (%)	0.44
Digestible lysine (%)	0.97
Digestible methionine + cystine (%)	0.57
Digestible threonine (%)	0.63
Digestible tryptophan (%)	0.21
Digestible arginine (%)	1.31
Digestible valine (%)	0.82

^†^ Fiber source;^ ‡^ Vitamin D 250 supplement: 25-OH-D3 (physiological precursor of the active hormone—vitamin D3) as the active ingredient; ^‡‡^ Content per kilogram of product: Folic Acid (15 mg/kg), Pantothenic Acid (400 mg/kg), BHT (100 mg/kg), Biotin (10 mg/kg), Calcium (200 mg/kg), Cobalt (5 mg/kg), Copper 1500 mg/kg), Choline (10 mg/kg), Chromium (5 mg/kg), Iron (1750 mg/kg), Phytase (12.50 FTU/kg), Phosphorus (82 g/kg), Iodine25 (mg/kg), L-carnitine (1261 mg/kg), Manganese (1000 mg/kg), Niacin (750 mg/kg), Selenium (7.5 mg/kg), Sodium (48.75 g/kg), Vitamin A (250,000 IU/kg), Vitamin B1 (38 mg/kg), Vitamin B12 (500 mcg/kg), Vitamin B2 (125 mg/kg), Vitamin B6 (25 mg/kg), Vitamin D3 (40,000 IU/kg), vitamin E (1500 IU/kg), vitamin K3 (50 mg/kg); ^§^ Contain 70% lysine; ^¶^ 96% guanidinoacetic acid (GAA) as creatine precursor; ^†††^ *Bacillus subtilis* and *B. licheniformes*; ^¶¶^ Live strain *Saccharomyces cerevisiae.*

**Table 2 animals-13-02764-t002:** Performance of lactating sows supplemented or not with phytogenic compounds.

	Control	Phytogenic Compound	CV (%)	*p* Value
*n*	49	44		
Lactation days (days)	20.55	20.80	8.95	0.541
Postpartum weight (kg)	250.36	250.87	12.41	0.987
Weaning weight (kg)	248.24	244.34	11.32	0.334
Mobilization (%)	0.85	2.60	384.60	0.201
Feed intake (kg/day)	5.036	5.080	12.93	0.981
Milk production (kg/day)	7.66	9.37	26.25	<0.001

The means differed at a level of 5% probability, measure by F-test.

**Table 3 animals-13-02764-t003:** Milk composition of sows supplemented or not with phytogenic compounds.

	Control	Phytogenic Compound	CV (%)	*p* Value
*n*	13	13		
Equalization				
Fat (%)	6.55	7.26	18.36	0.151
Crude Protein (%)	3.72	3.47	6.99	0.037
Lactose (%)	5.60	5.21	6.99	0.029
Non-fat solids (%)	9.93	9.63	7.65	0.395
Mineral fraction (%)	0.81	0.79	6.91	0.543
Total solids (%)	16.20	17.45	8.87	0.058
Weaning				
Fat (%)	5.67	6.31	20.82	0.467
Crude Protein (%) ^‡^	3.54	3.69	7.73	0.075
Lactose (%)	5.32	5.41	7.50	0.752
Non-fat solids (%)	9.73	10.03	7.38	0.494
Mineral fraction (%)	0.79	0.79	6.53	0.815
Total solids (%)	15.41	16.34	10.74	0.333

^‡^ Lactose at 20 days of lactation was used as a co-variable. The means differed at a level of 5% probability, confirmed by F-test.

**Table 4 animals-13-02764-t004:** Biochemical parameters of sows supplemented or not with phytogenic compounds.

	Control	Phytogenic Compound	CV (%)	*p* Value
*n*	13	13		
Equalization				
Total protein (g/dL)	5.70	5.80	7.65	0.599
Albumin (g/dL)	3.30	3.30	8.48	0.603
Globulin (g/dL)	2.40	2.50	18.48	0.857
Urea (mg/dL)	32.62	39.42	31.44	0.153
Creatinine (mg/dL)	1.31	1.20	22.61	0.755
Aspartate aminotransferase (U/L)	60.25	62.52	42.10	0.840
Creatine kinase (U/L)	3815.81	2240.65	80.25	0.121
Cholesterol (mg/dL)	45.56	41.72	30.35	0.503
Triglycerides (mg/dL)	38.15	35.18	36.53	0.477
Glucose (mg/dL)	80.08	74.50	13.59	0.184
Weaning				
Total protein (g/dL)	7.16	7.47	18.81	0.041
Albumin (g/dL)	3.18	3.34	12.33	0.416
Globulin (g/dL)	3.98	4.14	33.34	0.128
Urea (mg/dL)	41.22	48.80	21.25	0.043
Creatinine (mg/dL)	2.07	2.14	17.58	0.431
Aspartate aminotransferase (U/L)	49.00	54.09	32.11	0.541
Creatine kinase (U/L)	2290.80	2420.53	90.96	0.247
Cholesterol (mg/dL)	96.27	95.75	17.68	0.966
Triglycerides (mg/dL)	34.36	37.18	47.60	0.758
Glucose (mg/dL)	68.82	68.40	12.72	0.867

The means differed at a 5% probability level, confirmed by F-test.

**Table 5 animals-13-02764-t005:** Behavior of sows as a function of supplementation with phytogenic compounds during lactation.

Parameters	Control	Phytogenic Compound	CV (%)	*p* Value
Drinking water	1.45	2.40	309.68	0.002
Feed intake	2.05	2.65	330.30	0.226
Stereotypes	0.00	0.00	-	1.000
Inactive	60.22	63.38	37.85	0.044
Inactive alert	15.20	8.98	127.60	<0.001
Breastfeeding	21.08	22.58	95.39	0.009
Biting	0.00	0.00	-	1.000
Poking	0.00	0.00	-	1.000

The means differed at the 5% probability level, confirmed by Kruskal–Wallis test.

**Table 6 animals-13-02764-t006:** Performance and behavior of piglets as a function of supplementation with phytogenic compounds.

	Control	Phytogenic Compounds	CV (%)	*p* Value
*n*	49	44		
Equalization				
Number of piglets (*n*)	12.94	12.89	3.90	0.523
Average litter weight (kg)	19.97	20.21	16.54	0.728
Average weight of the piglets (kg)	1.545	1.567	16.29	0.655
Coefficient of variation (%)	18.72	19.69	26.00	0.358
Weaning				
Number of piglets (*n*)	11.51	11.73	11.64	0.62
Average litter weight (kg)	58.67	68.89	20.98	0.001
Average piglet weight (kg)	5.101	5.868	17.41	<0.001
Average daily weight gain (kg)	0.183	0.218	22.46	<0.001
Coefficient of variation (%)	21.34	17.91	32.05	0.010
Mortality (%)	10.98	8.92	101.74	0.502
Frequency of Diarrhea				
No diarrhea (%)	71.78	79.06	27.02	0.153
Diarrhea 1 (%)	12.85	11.87	89.38	0.956
Diarrhea 2 (%)	8.50	6.72	119.81	0.154
Diarrhea 3 (%)	6.87	2.35	163.11	0.004
Breastfeeding Behavior				
Duration (min)	7.70	9.37	49.90	<0.001
Feedings (*n*)	30.46	32.75	13.31	0.016
Interval (min)	38.85	34.19	48.36	<0.001

The means differed at the 5% probability level, confirmed by F-test.

## Data Availability

The data supporting this study cannot be publicly shared for ethical or privacy reasons, although may be shared upon reasonable request to the corresponding author if appropriate.

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
