# Peer review of "Phytogenic Bioactive Compounds in the Diet of Lactating Sows, Litter Performance, and Milk Characteristics"

_animals, 2023, doi:10.3390/ani13172764_

Round 1
Reviewer 1 Report
Manuscript ID: Animals-2476767
Phytogenic bioactive compounds in the diet of lactating sows, litter performance, and milk characteristics.
General comments: This topic is very interesting, and new data is always welcome.
This study includes the results of an experiment comparing the responses of some variables of lactating sows and their litters when they were supplemented with a blend of photobiotic bioactive compounds.
Introduction: I suggest including a few lines to indicate which is the gap in this topic,
L 66. Please check the expression “phytogenic bioactive compounds in the nutrition of sows in the lactation….”, to make it clear.
L 75: Please include the range of the parity.
L88. Please include de references for nutritional requirements
L90: The additive was added “on top”, but the dosage is given as a g/ton. I suggest informing the amount given.
L94: Why is mentioned the guaranteed level for anethole, but the concentration of other active compounds are missing? Could be useful to include those values.
Table 1. I respectfully suggest not including commercial names (consider editors’ rules). The composition of the vitamin and mineral premix could be included as a footnote.
L102. Why was the feed intake controlled? This could be a variable that could be affected by the tested additive.
L123: Why the sow´s behavior was evaluated? This is an important area but, it is not considered in your introduction or hypothesis.
L137: Please check: “the piglets of each herd”,
L141 – 145. Please check the stratification, this looks like is not correct or it is not clear.
L 147 – 149. I suggest adding the method to analyze diarrhea and behavior data.
Results
I respectfully suggest considering the order to present the results and discussion: since the aim of the paper is the performance of the sow and the litter, and the milk composition, these results should be the core of the results and the discussion. Also, it can improve the understanding of the paper.
Table 3: Was the lactose used as a covariable to compare values for crude protein at equalization time?
Discussion
L253 – 258: In my opinion, this is not relevant to the discussion.
L260-266: This paragraph involves many aspects, it is not clear.
L265 and L 266; L 268 – 271: There is no evidence on the present results of the effects of supplementation of “providing protein” since the feed intake was controlled.
Please check paragraphs: L260 to 266 and L289 to 296.
L282-L288: Please provide a reference for these sentences.
L306-307 Please explain what is “milk uniformity”
Author Response
Point-to-point response for reviewers
TITLE: Phytogenic bioactive compounds in the diet of lactating sows, litter performance, and milk characteristics.
Author: Gleyson Araújo dos Santos
Dear reviewers,
We thank you for the opportunity of submitting our manuscript once more (ID: animals-2476767).
First of all, we would like to thank you for your comments, criticisms and recommendation. The authors accepted all the reviewers' suggestion, to improve the writing and quality of this manuscript. All changes made were highlighted in the main file.
Revisor 1
This study includes the results of an experiment comparing the responses of some variables of lactating sows and their litters when they were supplemented with a blend of photobiotic bioactive compounds.
Introduction: I suggest including a few lines to indicate which is the gap in this topic,
Author: Thank you. We changed it according to your suggestion. The introduction was rewritten.
L 66. Please check the expression “phytogenic bioactive compounds in the nutrition of sows in the lactation….”, to make it clear.
Author: Thank you. We changed it according to your suggestion.
L 75: Please include the range of the parity.
Author: Thank you. We changed it according to your suggestion.
L88. Please include de references for nutritional requirements
Author: Thank you. We changed it according to your suggestion.
L90: The additive was added “on top”, but the dosage is given as a g/ton. I suggest informing the amount given.
Author: Thank you. We changed it according to your suggestion
L94: Why is mentioned the guaranteed level for anethole, but the concentration of other active compounds are missing? Could be useful to include those values.
Author: Thank you. We changed it according to your suggestion
Table 1. I respectfully suggest not including commercial names (consider editors’ rules). The composition of the vitamin and mineral premix could be included as a footnote.
Author: Thank you. We changed it according to your suggestion.
L102. Why was the feed intake controlled? This could be a variable that could be affected by the tested additive.
Author: Thank you. Feed intake was ad libitum, as described on "page 3 - line 115 – 119," and daily intake is described in Table 2.
L123: Why the sow´s behavior was evaluated? This is an important area but, it is not considered in your introduction or hypothesis.
Author: Thank you. The introduction was rewritten. This information was reported on page 2 - line 49.
L137: Please check: “the piglets of each herd”,
Author: Thank you. We changed it according to your suggestion
L141 – 145. Please check the stratification, this looks like is not correct or it is not clear.
Author: Thank you. We changed it according to your suggestion
L 147 – 149. I suggest adding the method to analyze diarrhea and behavior data.
Author: Thank you. We changed it according to your suggestion.
Results
I respectfully suggest considering the order to present the results and discussion: since the aim of the paper is the performance of the sow and the litter, and the milk composition, these results should be the core of the results and the discussion. Also, it can improve the understanding of the paper
Author: Thank you. We changed it according to your suggestion. the structure for presenting the results and discussions was changed and rewritten
Table 3: Was the lactose used as a covariable to compare values for crude protein at equalization time?
Author: Thank you. data were subjected to correlation analysis to determine the relationship between variables. The criterion for using the covariate was based on the correlation between the parameters and the days of milk collection (at the time of equalization and at weaning), and when the correlation was highly significant, values greater than 0.87, the parameter was selected as covariable. A similar methodology was adopted in the article “Moreira, R.H.R., Perez Palencia, J.Y., Moita, V.H.C., Caputo, L.S.S., Saraiva, A., Andretta, I., Ferreira, R.A. and de Abreu, M.L.T., 2020. Variability of piglet birth weights: A systematic review and meta‐analysis. Journal of animal physiology and animal nutrition, 104(2), pp.657-666. DOI: 10.1111/jpn.13264”
Discussion
L253 – 258: In my opinion, this is not relevant to the discussion.
Author: Thank you. We changed it according to your suggestion and the sentence was withdrawn.
L260-266: This paragraph involves many aspects, it is not clear.
Author: Thank you. We changed it according to your suggestion. It was rewritten.
L265 and L 266; L 268 – 271: There is no evidence on the present results of the effects of supplementation of “providing protein” since the feed intake was controlled.
Author: Thank you. We changed it according to your suggestion and the sentence was withdrawn.
Please check paragraphs: L260 to 266 and L289 to 296.
Author: Thank you. We changed it according to your suggestion and the sentence was withdrawn.
L282-L288: Please provide a reference for these sentences.
Author: Thank you. We changed it according to your suggestion, new references have been added.
L306-307 Please explain what is “milk uniformity”
Author: Thank you. Got corrected.

Reviewer 2 Report
Comments to the Author
The manuscript entitled" Phytogenic bioactive compounds in the diet of lactating sows, litter performance, and milk characteristics” is well organized. The idea and design of this study is well although it is not clear.
However, I suggest the authors should address the following comments for further consideration of the manuscript for publication.
I suggested improve the background of the study.
The authors should give expansions for the abbreviations used in the abstract for the first time appear and followed by the short form can be used. The same has to be followed for the remaining part of the manuscript.
The introduction must be improved with more relevant information about the core plot of the manuscript.
In the introduction section, the references are very old. Few recent references must be included for better promotion of this manuscript.
clarify whether the phytogenic bioactive compounds in the diet of lactating sows, litter performance, and milk characteristics was formulated or top dressed in the diet. Also indicate whether the diet was in mash or pellet form, if latter what were the conditions of processing. Could you explain the preparation?
The authors should clearly define the protocol use in this experiment and this is applied to the results part.
Results: all descriptions of the results should be reworked. The following parts are required, 1) the comparison of the significant difference among the values of treatments; The result descriptions should be presented in subsections based on the type of indicators.
The sows conditions should be described in more detail. In particular, it is recommended to report the genetic type, the dimensions of the pens, the presence of enrichment materials, and the final live weights.
The software package should be included in the materials and method section and the statistical analyses were poorly explained. Please rewrite
In discussion, the authors have mentioned many references, but in many part it is not correlated with the present findings and it is encouraged to includes how the present findings results are observed for the better outcome of the manuscript.
Conclusion: write the conclusion again and be more concise to the major findings and suggestions.
I find some formal errors in the manuscript, above all it is necessary to check the correctness of the literary references in the text according to the list of references (dates of publication) and also the correctness of writing the references in the text (dots, commas, spaces...). Not all abbreviations are explained in the text.
Grammar, punctuation, sentence structure, use of past/present tense and the use of plurals within the manuscript are quite consistent. Thus, it may be to the authors benefit to make use of a professional editing service to perfect the manuscript.
The format of references needs to be standardized.
Grammar, punctuation, sentence structure, use of past/present tense and the use of plurals within the manuscript are quite consistent. Thus, it may be to the authors benefit to make use of a professional editing service to perfect the manuscript.
Author Response
Point-to-point response for reviewers
TITLE: Phytogenic bioactive compounds in the diet of lactating sows, litter performance, and milk characteristics.
Author: Gleyson Araújo dos Santos
Dear reviewers,
We thank you for the opportunity of submitting our manuscript once more (ID: animals-2476767).
First of all, we would like to thank you for your comments, criticisms and recommendation. The authors accepted all the reviewers' suggestion, to improve the writing and quality of this manuscript. All changes made were highlighted in the main file.
Revisor 2
The manuscript entitled" Phytogenic bioactive compounds in the diet of lactating sows, litter performance, and milk characteristics” is well organized. The idea and design of this study is well although it is not clear. However, I suggest the authors should address the following comments for further consideration of the manuscript for publication.
I suggested improve the background of the study.
The authors should give expansions for the abbreviations used in the abstract for the first time appear and followed by the short form can be used. The same has to be followed for the remaining part of the manuscript.
Author: Thank you. We changed it according to your suggestion. It was rewritten.
The introduction must be improved with more relevant information about the core plot of the manuscript.
Author: Thank you. We changed it according to your suggestion. It was rewritten.
In the introduction section, the references are very old. Few recent references must be included for better promotion of this manuscript.
Author: Thank you. We changed it according to your suggestion. The introduction should be enhanced with more relevant information about the central plot of the manuscript.
Clarify whether the phytogenic bioactive compounds in the diet of lactating sows, litter performance, and milk characteristics was formulated or top dressed in the diet. Also indicate whether the diet was in mash or pellet form, if latter what were the conditions of processing. Could you explain the preparation?
Author: Thank you. All information has been included.
The authors should clearly define the protocol use in this experiment and this is applied to the results part.
Results: all descriptions of the results should be reworked. The following parts are required, 1) the comparison of the significant difference among the values of treatments; The result descriptions should be presented in subsections based on the type of indicators.
Author: Thank you. We changed it according to your suggestion
The sows conditions should be described in more detail. In particular, it is recommended to report the genetic type, the dimensions of the pens, the presence of enrichment materials, and the final live weights.
Author: Thank you. All information has been included. The average weight of the sows at the beginning and end of the study was described in Table 2.
The software package should be included in the materials and method section and the statistical analyses were poorly explained. Please rewrite
Author: Thank you. We changed it according to your suggestion, it was rewritten.
In discussion, the authors have mentioned many references, but in many part it is not correlated with the present findings and it is encouraged to includes how the present findings results are observed for the better outcome of the manuscript.
Author: Thank you. All information has been included.
Conclusion: write the conclusion again and be more concise to the major findings and suggestions.
I find some formal errors in the manuscript, above all it is necessary to check the correctness of the literary references in the text according to the list of references (dates of publication) and also the correctness of writing the references in the text (dots, commas, spaces...). Not all abbreviations are explained in the text.
Author: Thank you. All information has been included.
Grammar, punctuation, sentence structure, use of past/present tense and the use of plurals within the manuscript are quite consistent. Thus, it may be to the authors benefit to make use of a professional editing service to perfect the manuscript.
The format of references needs to be standardized.
Author: Thank you. A language correction has been made
Kind regards,
Authors

Round 2
Reviewer 2 Report
Authors provided proper improvement of their text according to reviewers' comments. Therefore, the manuscript can be considered for further publication.
Best regards